# Machine learning with electroencephalography features for precise diagnosis of depression subtypes

**Maria Zelenina**[1]                                                MARIE.ZELENINA@GMAIL.COM

**Diana Maria Pinto Prata**[1,2,3]                                    DIANA.PRATA@KCL.AC.UK

[1] *Instituto de Biofisica e Engenharia Biomedica, Faculdade de Ciencias da Universidade de Lisboa, Portugal*

[2] *Institute of Psychiatry, Psychology and Neuroscience, King's College London, UK*

[3] *Centre for Psychological Research and Social Intervention (CIS), University Institute of Lisbon (ISCTE-IUL), Portugal*

## Abstract

Depression is a common psychiatric disorder, which causes significant patient distress. Bipolar disorder is characterized by mood fluctuations between depression and mania. Unipolar and bipolar depression can be easily confused because of similar symptom profiles, but their adequate treatment plans are different. Therefore, a precise data-driven diagnosis is essential for successful treatment. In order to aid diagnosis, research applied machine learning to brain imaging data, in particular to electroencephalography (EEG), with accuracies reaching 99.5% (unipolar vs. healthy) or 85% (bipolar vs. healthy). However, these results arise from small training sets, without validation on independent data, and thus have a high risk of inflated accuracies due to data over-fitting. We propose to use a bigger corpus of realistic clinical data for training and testing and improve classification with microstates features, which can assess the function of large-scale brain networks.

**Keywords:** Neuroscience, machine learning, EEG, depression, bipolar.

## 1. Introduction

Depressive disorders are the 4th leading cause of disability worldwide (WHO, 2016). Successful treatment of depression depends on a precise diagnosis, so it is important to develop classification tools that allow not only to distinguish depressed patients from healthy controls, but also to identify subtypes of depression. Especially urgent is the differentiation between unipolar and bipolar depression because they require different treatment plans, and misdiagnosis can delay recovery. It is, however, a demanding task, because unipolar and bipolar depression share a similar symptom profile (Hui et al., 2018). This makes a data-driven computer-aided precise diagnosis of depression subtypes based on biomarkers, rather than symptoms, an attractive approach. Most attempts to aid in diagnosis of depression with machine learning tools applied to neuroimaging data have been centred around Magnetic Resonance Imaging (MRI) (Gao et al., 2018). MRI benefits from advantages in spatial composition of the data, but has disadvantages in terms of low temporal composition, high cost, and requirement of more extensively trained research personnel, compared to EEG. Finally, fMRI measures brain activity indirectly, using blood oxygenation measures

as a proxy of brain activity, while EEG measures the electrical activity of the brain directly. This makes EEG an attractive brain imaging tool for neuropsychiatric disorders.

### 1.1. Computer-aided diagnosis of unipolar depression with EEG

In the past 20 years, the classification of unipolar depression based on EEG features attracted significant research effort (Acharya et al., 2018, 2015; Ahmadlou et al., 2012, 2013; Bairy et al., 2017; Cukic et al., 2018; Faust et al., 2014; Hosseinifard et al., 2013; Knott et al., 2001; Liao et al., 2017; Mumtaz et al., 2017, 2018; Puthankattil and Joseph, 2012). Notably, some of these attempts reach an accuracy of up to 99.5% (Faust et al., 2014). However, all those results were achieved on small datasets (22 to 90 recordings) obtained in non-naturalistic highly controlled research settings and were not replicated in independent clinical samples, which raises doubts in scalability of results and their translatability to realistic clinical settings.

### 1.2. Computer-aided diagnosis of bipolar vs. unipolar depression with EEG

So far, there were few attempts to differentiate unipolar from bipolar depression with machine learning tools applied to EEG data (Khodayari-Rostamabad et al., 2010; Erguzel et al., 2015, 2016). A maximum likelihood approach based on the combination of factor analysis models achieved an average correct diagnosis rate (prediction accuracy) of around 85% (Khodayari-Rostamabad et al., 2010). A support vector machine classifier achieved a prediction accuracy of 80.19% (Erguzel et al., 2015). Finally, an artificial neural network achieved a prediction accuracy of 83.87% (Erguzel et al., 2016). All those attempts suffered from the same problems with scalability & translatability as unipolar classification research.

Given the therapeutic importance of precise diagnosis of bipolar vs. unipolar depression, further efforts are required to improve the classification precision. We suggest contributing to those efforts by adding microstates features, which can tap into the function of large-scale brain networks, and have shown great potential as biomarkers in neuropsychiatry research.

## 2. Proposed methods

### 2.1. Data

We intend to use a large EEG sample, including over 650 unipolar and over 250 bipolar depression patients and healthy controls, from multiple hospitals hospitals, for training and testing of our classifier. 90% of this data will be used for training and testing of our model. To validate our results, we will set aside a smaller sample (10%) of the data, which will consist of recordings coming from a different hospital setting than the data used for training and testing.

### 2.2. Features

Inspired by previous research (Erguzel et al., 2015, 2016; Khodayari-Rostamabad et al., 2010; Tas et al., 2015; Cai et al., 2018; Khaleghi et al., 2015; Cukic et al., 2018; Hosseinifard et al., 2013; Strik et al., 1995), we propose the following features:

1. Band features: cordance, spectral coherence, absolute power.

2. Time domain features: mean, variance, kurtosis, skewness, Hjorth parameters.

3. Non-linear features: entropy, C0-complezity, fractal dimension.

4. Microstates features: duration, occurence, contribution, transition probabilities. Research has not used microstates to classify unipolar vs. bipolar depression, although Strik et al. suggested altered microstate patters in depression (Strik et al., 1995).

### 2.3. Machine learning

#### 2.3.1. Feature selection

We propose to use a wrapper feature selection method developed specifically for EEG data (Hossain et al., 2014). It implements a backward search strategy, which starts with the full feature set and iteratively removes features using correlation criteria. Because microstates have not been previously used to differentiate unipolar and bipolar depression, we intend to research microstates features separately as an additional step, with the use of analysis of variance (ANOVA), confusion matrices and single variable classifiers.

#### 2.3.2. Classifiers

A recent review (Yannick et al., 2019) highlighted advantages of deep learning in applicability to EEG data because of its capacity to learn good feature representations from raw data. They discuss that deep neural networks, in particular convolutional networks, could learn features from raw or minimally preprocessed data. As such, we propose to experiment with fully connected, convolutional, and recurrent neural networks. However, we still intend to preprocess the data and perform feature selection as described above, and compare performances of pipelines with manual preprocessing and wrapper feature selection to no or minimal preprocessing and automatic feature learning with neural networks.

Additionally, we propose to experiment classifiers such as Logistic Regression, Linear Discriminant Analysis, K-Nearest Neighbors, Decision Trees, including the recent highly popular LightGBM (Ke et al., 2017), Naive Bayes, and Support Vector Machines. The final choice of the algorithm will be done empirically. For training and testing, we will split the data with the 80-20 ratio. To maximize the amount of data available for training, we will use leave-one-out cross-validation.

#### 2.3.3. Evaluation techniques

We will evaluate our model using AUC-ROC (Area Under The Curve - Receiver Operating Characteristics) analysis. It is an aggregate measure of performance across all possible classification thresholds, which can be interpreted as the probability that the model ranks a random positive example more highly than a random negative example. It is scale-invariant and measures how well predictions are ranked, rather than their absolute values. Also, it is classification-threshold-invariant and measures the quality of the models predictions irrespective of the classification threshold. AUC-ROC values range from 0 (worst) to 1 (best). Additionally, we will use the classification accuracy score, which is the ratio of number of correct predictions to the total number of input samples. We will consider precision accuracy higher than human diagnostic error a successful validation procedure.

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
