# OpenReview forum: "Machine learning with electroencephalography features for precise diagnosis of depression subtypes"
_MIDL.io/2019/Conference/Abstract — MIDL Abstract 2019_

### Official Review · AnonReviewer2 · 2019-04-29
**Interesting idea but results are unfortunately not reported**

**Rating:** 2
**Confidence:** 3

**Review:**

The paper proposes and interesting approach to process EEG signals to differentiate between different subtypes of depression.

Although the method could in principle work and although it would surely have an important impact on this particular task, no experimentation seems to have been performed yet. The authors discuss the results achieved by other state of the art methods, but don't seem to discuss their own.
At other venues, this is perfectly acceptable when submitting an abstract. I am not sure if this is acceptable at MIDL, as I have always seen abstracts that report some quantitative or (at least) qualitative results.

I would recommend that the authors submit a new paper once they obtain the results their experimental evaluation. For now, unfortunately, the abstract is in my opinion not acceptable because the validity of the method cannot be assessed without experimental evaluation.

---

### Official Review · AnonReviewer1 · 2019-05-01
**"use a bigger corpus of realistic clinical data for training and testing" is good progress**

**Rating:** 4
**Confidence:** 2

**Review:**

The paper is detailed enough and sufficiently convincing. The problem is interesting and important. The work's design protocol on addressing a larger study population and the generality on the proposed approach is good.

---

### Decision · Program_Chairs · 2019-05-06
**Acceptance Decision**

Accept